# Architecture and dynamics of the autophagic phosphatidylinositol 3-kinase complex

**Sulochanadevi Baskaran[1,2†], Lars-Anders Carlson[1,2†], Goran Stjepanovic[1,2†], Lindsey N Young[1,2], Do Jin Kim[1,2], Patricia Grob[1,2,5], Robin E Stanley[3‡], Eva Nogales[1,2,4,5]\*, James H Hurley[1,2,4]\***

[1]Department of Molecular and Cell Biology, University of California, Berkeley, Berkeley, United States; [2]California Institute for Quantitative Biosciences, University of California, Berkeley, Berkeley, United States; [3]Laboratory of Molecular Biology, National Institute of Diabetes and Digestive and Kidney Diseases, National Institutes of Health, Bethesda, United States; [4]Life Sciences Division, Lawrence Berkeley National Laboratory, Berkeley, United States; [5]Howard Hughes Medical Institute, University of California, Berkeley, Berkeley, United States

**\*For correspondence:**
enogales@lbl.gov (EN);
jimhurley@berkeley.edu (JHH)

[†]These authors contributed equally to this work

**Present address:** [‡]Laboratory of Signal Transduction, National Institute of Environmental Health Sciences, Department of Health and Human Services, National Institutes of Health, Bethesda, United States

**Competing interests:** The authors declare that no competing interests exist.

**Abstract** The class III phosphatidylinositol 3-kinase complex I (PI3KC3-C1) that functions in early autophagy consists of the lipid kinase VPS34, the scaffolding protein VPS15, the tumor suppressor BECN1, and the autophagy-specific subunit ATG14. The structure of the ATG14-containing PI3KC3-C1 was determined by single-particle EM, revealing a V-shaped architecture. All of the ordered domains of VPS34, VPS15, and BECN1 were mapped by MBP tagging. The dynamics of the complex were defined using hydrogen–deuterium exchange, revealing a novel 20-residue ordered region C-terminal to the VPS34 C2 domain. VPS15 organizes the complex and serves as a bridge between VPS34 and the ATG14:BECN1 subcomplex. Dynamic transitions occur in which the lipid kinase domain is ejected from the complex and VPS15 pivots at the base of the V. The N-terminus of BECN1, the target for signaling inputs, resides near the pivot point. These observations provide a framework for understanding the allosteric regulation of lipid kinase activity.

## Introduction

Macroautophagy (hereafter, 'autophagy') is a conserved eukaryotic pathway for cellular self-preservation through cellular self-consumption (*Mizushima et al., 2011*; *Rubinsztein et al., 2012b*; *Reggiori and Klionsky, 2013*; *Green and Levine, 2014*). The most ancient role for autophagy is surviving during starvation. Bulk cytosol is captured in a growing double membrane structure termed the phagophore. Upon sealing, the resulting double membrane vesicle is known as an autophagosome. The autophagosome fuses with the lysosome or vacuole, resulting in the degradation of its contents and the recycling of biosynthetic precursors by export across the lysosomal membrane. In higher eukaryotes, autophagy has acquired many additional roles in cellular protection. For example, the core autophagy protein BECN1 (Beclin 1) is a tumor suppressor. *BECN1* is deleted in 40–75% of human breast, ovarian, and prostate cancers (*Liang et al., 1999*). Autophagy is also thought, under most conditions, to protect cells from the accumulation of pathogenic inclusions that can lead to Huntington's, Parkinson's, and other neurodegenerative diseases (*Rubinsztein et al., 2012a*; *Nixon, 2013*).

During the initiation of autophagy, an autophagy-specific form of the class III phosphatidylinositol 3-kinase complex (PI3KC3-C1) is activated and recruited to the site of phagophore nucleation. Class III PI3Ks synthesize phosphatidylinositol 3-phosphate (PI(3)P) from phosphatidylinositol (PI), as compared

**eLife digest** To survive starvation and other hard times, cells have developed a unique recycling strategy: they can scavenge the resources they need from within the cell itself. To do this, the cell forms a double-layered envelope around particular sections of the cell to seal them off from the rest. Then, the contents of the envelope are taken apart and the resulting raw materials are sent elsewhere in the cell where they can be used as required. This process is called autophagy.

In more complex organisms like humans, autophagy can have additional roles. One of the key proteins involved in autophagy—called BECN1—suppresses the growth of tumors, and the gene that makes BECN1 is missing in 40–70% of human breast, ovarian, and prostate cancers. Autophagy may also help to prevent Huntington's disease and other similar conditions by stopping disease-causing proteins or broken cell parts from building up inside brain cells.

The BECN1 protein does not work alone. Instead, it becomes part of a group, or 'complex', of several proteins that are required to form the envelope made during autophagy. However, the three-dimensional structure of the protein complex is unclear.

Baskaran et al. used electron microscopy and other techniques to investigate this structure and found that the complex forms a V shape with two arms, which is held together by its largest protein, VPS15. This protein also acts as a bridge between BECN1 and another protein that is a target for new cancer drugs, called VPS34.

Next, Baskaran et al. used a different set of techniques to determine how the complex moves. This revealed that many of the connections between proteins in the complex are flexible. However, one of the arms is inflexible and this limits the ability of the VPS34 protein to move. Understanding this structural constraint may help us to design drugs that are able to target the protein complex more efficiently.

to the class I and II PI3Ks that generate PI(3,4,5)$P_3$ and PI(3,4)$P_2$, respectively (*Backer, 2008*). In yeast, where the complex was first characterized, there are two PI 3-kinase complexes, known as complexes I and II, with clear-cut distinctions in their functions (*Kihara et al., 2001*). Both yeast complexes contain the core subunits Vps34, Vps15, and Atg6. Vps34 contains the catalytic domain responsible for lipid kinase activity, as well as a putative lipid-binding C2 domain and a helical domain (*Schu et al., 1993*; *Miller et al., 2010*). Vps15 is a large (150 kDa) protein essential for the activity of the complex (*Stack et al., 1993*). Vps15 contains a protein kinase domain whose function and possible substrates are uncertain, as well as HEAT and WD40 repeats. Atg6 is the yeast ortholog of the human tumor suppressor BECN1 and contains an intrinsically disordered region, a coiled coil, and a BARA domain. Yeast complex I further contains the coiled coil subunit Atg14 and is the complex that is involved in autophagy initiation. Yeast complex II contains instead another coiled coil protein, Vps38, and functions in endosome maturation.

The yeast complexes are prototypes of two human PI 3-kinase complexes (*Volinia et al., 1995*), which both contain the common core subunits VPS34 (also known as PI3KC3), VPS15, and BECN1 (*Backer, 2008*; *Funderburk et al., 2010*; *He and Levine, 2010*; *Wirth et al., 2013*) (*Figure 1A*). The human cognate of PI3KC3 complex I contains ATG14 (also known as ATG14L or BARKOR) (*Itakura et al., 2008*; *Sun et al., 2008*; *Matsunaga et al., 2009*; *Zhong et al., 2009*). We will refer to this complex as PI3KC3-C1. ATG14 is specifically recruited to sites of autophagosome initiation (*Matsunaga et al., 2010*; *Fogel et al., 2013*; *Graef et al., 2013*; *Hamasaki et al., 2013*; *Wirth et al., 2013*; *Ge et al., 2014*). A second human PI3KC3 contains, in place of ATG14, the UV resistance-associated gene product, UVRAG. We will refer to this complex as PI3KC3-C2. PI3KC3-C2 has been proposed to function at later stages in autophagy (*Liang et al., 2006*) but is not currently thought to participate in the initiation of autophagosome biogenesis. The overall objective of this study was to define the architecture of the two human PI3KC3 complexes. One aspect of the larger goal was to ascertain whether there are gross structural differences between PI3KC3-C1 and -C2 that might affect their relative ability to target the ER and initiate autophagy.

The pivotal role of PI3KC3 in the basic mechanism of autophagy induction and the potential importance of PI3KC3 modulators in treating cancer, neurodegenerative, and other diseases are driving interest in the structure of the complex. To date, crystal structures have been obtained for the catalytic

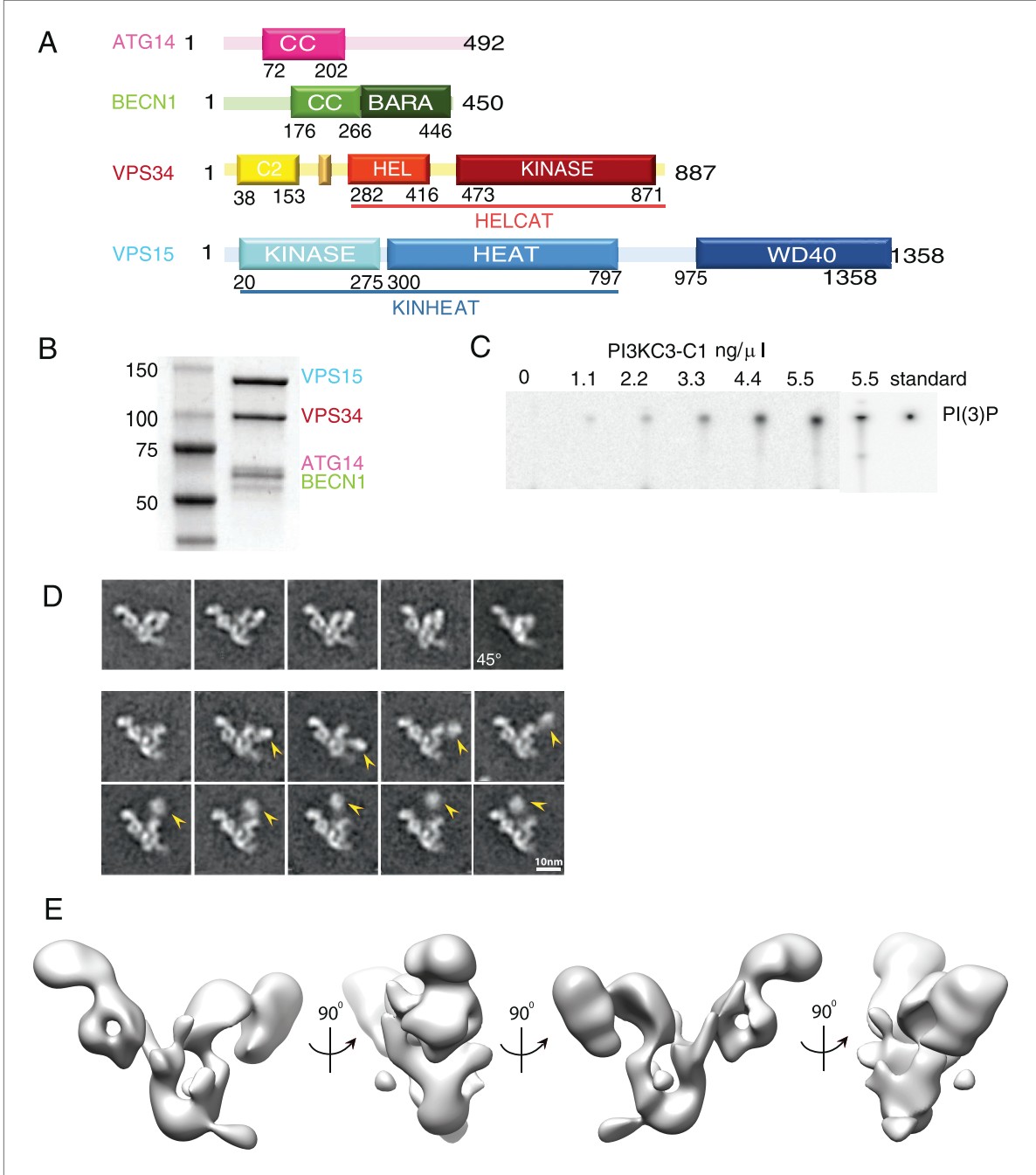

**Figure 1.** Reconstitution and 3D reconstruction of the PI3KC3-C1 complex. (**A**) Domain structures of the four subunits of PI3KC3-C1. The small unlabeled yellow box between VPS34 C2 and helical domains is the CHIL motif, described below. (**B**) Purification of PI3KC3-C1. Coomassie-stained SDS-PAGE gel of purified PI3KC3-C1. (**C**) Thin-layer chromatography of radiolabeled PI(3)P generated by PI3KC3-C1 from PI and [γ-32 P] ATP. (**D**) Reference-free class averages of PI3KC3-C1, each containing ~200 particles. The first row represents particles that were selected for the 3D reconstruction. The lower two rows represent particles excluded from the 3D reconstruction, the arrowhead indicating the position of a density dislodged from the main part of the complex. All class averages shown are calculated from data acquired on untilted grids, except the top right class average which is from 45° tilt. (**E**) 3D recon-struction calculated from ~39,000 particles in four orientations and displayed at a threshold determined from subsequent docking analyses.

core and helical domains of *Drosophila* Vps34 (*Miller et al., 2010*), the WD40 propeller domain of yeast Vps15 (*Heenan et al., 2009*), the coiled coil domain of human BECN1 (*Li et al., 2012*), and the BARA domains of yeast and human Atg6/BECN1 (*Huang et al., 2012*; *Noda et al., 2012*). Structures of homologs of the Vps34 C2 domain and the Vps15 HEAT repeats and protein kinase domains are

available. These various regions have to work together in a single complex in autophagy initiation. Currently, there are essentially no data on the structure of the 361.8 kDa quaternary assembly of these subunits with each other in PI3KC3-C1 (*Hurley and Schulman, 2014*). We view this information as essential, if we are to begin to understand how autophagy is initiated and how it is regulated and to address the potential for therapeutic modulation of PI3KC3-C1.

To this end, we reconstituted active human PI3KC3-C1 and PI3KC3-C2 by co-expression of all four subunits and imaged the complex by electron microscopy (EM). The complexes are elongated, loosely connected, and dynamic; yet by careful selection of class averages and Bayesian data processing, we were able to generate a three-dimensional reconstruction of PI3KC3-C1 at 28-Å resolution. We have mapped the positions of the various domains of the subunits by tagging with maltose binding protein (MBP). Insight into large-scale conformational fluctuations of PI3KC3-C1 was obtained from EM, while local dynamics was mapped by hydrogen–deuterium exchange. Taken together, the data led us to a model for the subunit architecture and dynamics of PI3KC3-C1. The main architectural principles also hold for the PI3KC3-C2 complex. The structural model suggests that the VPS15 kinase domain acts as a latch to regulate PI 3-kinase activity in both the PI3KC3-C1 and -C2.

## Results

### Reconstitution and imaging of PI3KC3-C1

In order to generate sufficient quantities of compositionally homogeneous PI3KC3-C1, synthetic DNA constructs encoding VPS34, VPS15, BECN1, and ATG14 were co-transfected in HEK293 cells. The resulting complex contained all four subunits at apparently equal stoichiometry (*Figure 1B*) and the subunits co-migrated as a single peak on gel filtration chromatography (*Figure 2A*). The material was enzymatically active as judged by ATP hydrolysis in the presence of PI (*Figure 2B*) and by the formation of PI(3)P from PI as assessed by thin layer chromatography (*Figure 1C*). ATP hydrolysis by PI3KC3-C1 was essentially completely blocked by the PI 3-kinase inhibitor wortmannin (*Figure 2B*), confirming that the ATPase activity was due to the lipid kinase domain and not either the VPS15 kinase domain or contaminants.

PI3KC3-C1 samples were stained with uranyl formate and imaged by EM (*Figure 3A*). Two-dimensional class averages for PI3KC3-C1 are shown in *Figure 1D*. The class averages revealed a V-shaped particle with two arms of approximately 20 nm in length. There was a strongly preferred orientation with the V view visible face-on in projection (*Figure 3B*). Edge-on views were less common. An ab initio 3D reconstruction was generated by the random conical tilt (RCT) method (*Radermacher et al., 1987*) and refined using particles acquired at 0°, 30°, and 45°. Class averages from ~160,000 particles were culled of aggregates, contaminants, lower resolution classes, and those with sparsely populated conformations, leading to a final set of ~40,000 high-quality particles. The final reconstruction (*Figure 1E*)

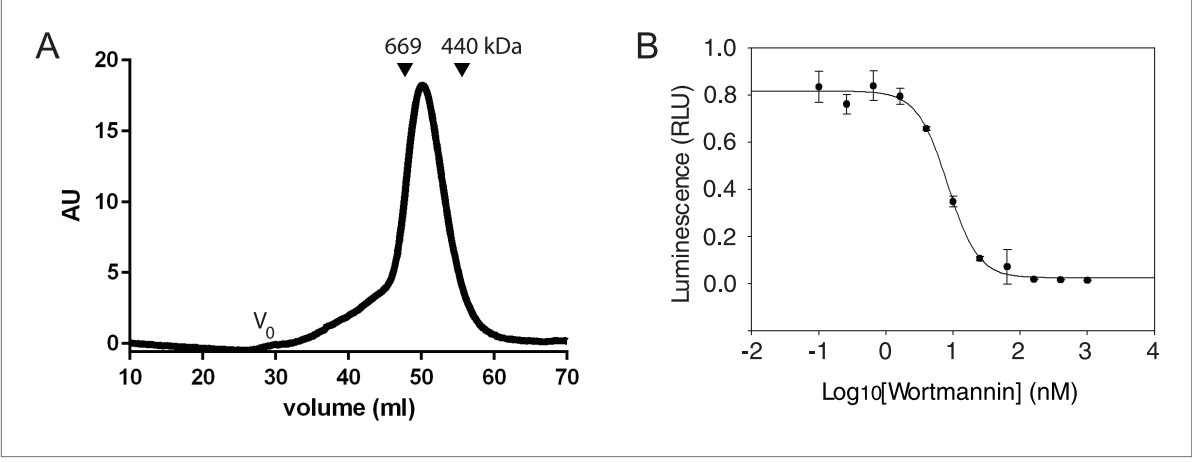

**Figure 2**. Characterization of the PI3KC3-C1 complex. (**A**) Size-exclusion chromatography of PI3KC3-C1 showing that the complex elutes as a single peak well separated from the void volume ($V_o$). (**B**) ATP hydrolysis by PI3KC3-C1 is inhibited by wortmannin. RLU, relative luminescence units × $10^6$.

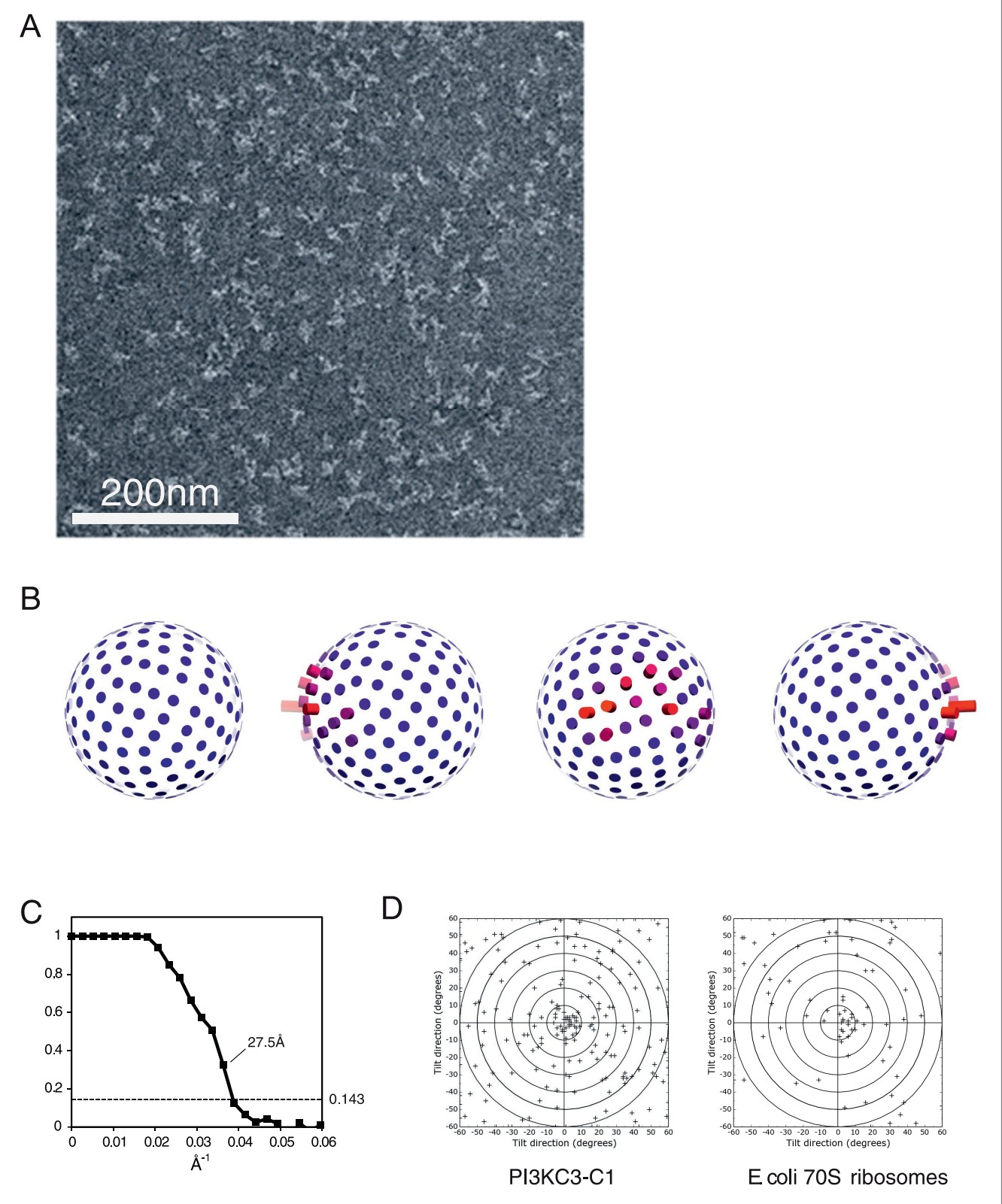

**Figure 3**. Electron microscopy of PI3KC3-C1. (**A**) Representative unprocessed micrograph of PI3KC3-C1. (**B**) Euler angle distribution of particles used for the 3D reconstruction shown in *Figure 1E*. The views are the same as those shown in *Figure 1E*. Long/red bars mean many particles contributing to the view along that bar. (**C**) Gold-standard Fourier shell correlation (FSC) plot of the 3D reconstruction shown in *Figure 1E*. (**D**) Tilt-pair analysis to validate
*Figure 3. Continued on next page*

Figure 3. Continued

the absolute hand of the 3D reconstruction shown in *Figure 1E*. Plots calculated from image pairs acquired at 0° and 15° for PI3KC3-C1 (left panel) and as control for *E. coli* 70S ribosomes (right panel). The ribosome plot was calculated using a previously published map deposited in EMDB with accession code EMD-1849 (*Agirrezabala et al., 2011*).

was calculated using a single 3D class in RELION (*Scheres, 2012*), with the initial RCT structure low-pass filtered to 80 Å as the initial reference. This reconstruction has a resolution of 28 Å according to the gold standard FSC criterion as implemented in RELION (*Figure 3C*) and was used for docking of known crystal structures or homology models of domains. The reconstructed density shows some distortion in the direction normal to the plane of the preferred particle orientation, due to the missing data for orientations at >45° tilt. The chirality of the structure was validated by tilt-pair analysis (*Rosenthal and Henderson, 2003*) with image pairs collected at 0° and 15° (*Figure 3D*).

## Subunit identification using tagged PI3KC3-C1 complexes

The 28 Å reconstruction described above lacked the detail, on its own, to unambiguously place most of the structures of the various subunit domains. The exceptions were the donut-shaped region corresponding to the WD40 domain of VPS15 and the arch-shaped density at the base of the V corresponding to the HEAT repeat of the same subunit. Therefore, we generated and purified a series of N- and C-terminal MBP fusion constructs to map the domains using EM (*Figure 4A,B*). Both N- and C-terminal MBP fusions of VPS34 and BECN1 and the N-terminal MBP fusions of ATG14 and VPS15 were successfully purified and imaged.

With respect to the canonical 'V' view of the complex, the VPS34 kinase subunit is at the tip of the right-hand arm of the V. Its N-terminus is near the crotch of the V. In turn, BECN1 is aligned with the left arm of the complex. Its N-terminus is near the junction of the V and its C-terminus at the tip of the left arm. The N-terminus of scaffolding VPS15 subunit is located near the C-terminus of VPS34 in the right arm. Finally, the N-terminus of ATG14 is near that of BECN1. Strikingly, this suggests that the BECN1 and ATG14 coiled coils must be parallel to each other, in contrast to the antiparallel coiled coil described for the BECN1 homodimer (*Li et al., 2012*).

## Three-dimensional subunit architecture of PI3KC3-C1

Given the added guidance provided by MBP tags, we docked the known crystal structures or homology models of the structured domains of VPS34, VPS15, and BECN1 on to the density and the docked volume accounted for 65% of the total density. (*Figure 5A,C*). The helical and catalytic domains of VPS34 are integrated into a single structural unit referred to as the 'HELCAT' region (*Miller et al., 2010*) (*Figure 1A*), which matches in size and shape the density identified near the C-terminal MBP tag in this subunit (*Figures 4B and 5C*). Thus, the HELCAT region was positioned such that its C-terminal helix was proximal to the position of the C-terminal MBP tag (*Figures 4B and 5C*). A density feature matching the size and shape of the C2 domain is located between the HELCAT and the N-terminal MBP tag (*Figures 4B and 5C*). There is no available crystal structure for VPS34-C2. However, a model based on the closely related structure of the C2 domain of PI3KC2α (*Liu et al., 2006*) was docked into this density, and a reasonable fit was obtained (*Figure 5C*).

The crystal structure of the BARA domain (also sometimes referred to as the 'ECD') of human BECN1 (*Huang et al., 2012*) was unambiguously identified as a match to the size and shape of the only density feature near the BECN1 C-terminal MBP tag (*Figures 4B and 5C*). The remaining large density features were assigned to VPS15, beginning with the N-terminally tagged kinase domain. A homology model for the VPS15 kinase catalytic domain was derived from the structure of protein kinase A (PKA) (*Knighton et al., 1991*). The model was docked into a major density feature adjoining the position of the tag and matching its size and shape (*Figures 4B and 5C*). A model for the 498-residue HEAT repeat domain of VPS15 was generated by threading the corresponding sequence onto the backbone of the HEAT repeat portion of the exportin subunit Cse1 (*Matsuura and Stewart, 2004*) and provided an excellent match to the arch-shaped density adjoining the C-terminus of the VPS15 kinase domain (*Figures 4B and 5C*). Indeed, the kinase and HEAT repeat domains of VPS15 seem to be fused into a single contiguous unit. The most C-terminal domain of VPS15, the WD40 propeller region, is connected to the HEAT by a 180-residue linker. This propeller was assigned to the last unaccounted for density feature, located next to the BECN1 BARA domain, which has a donut-like shape that matches

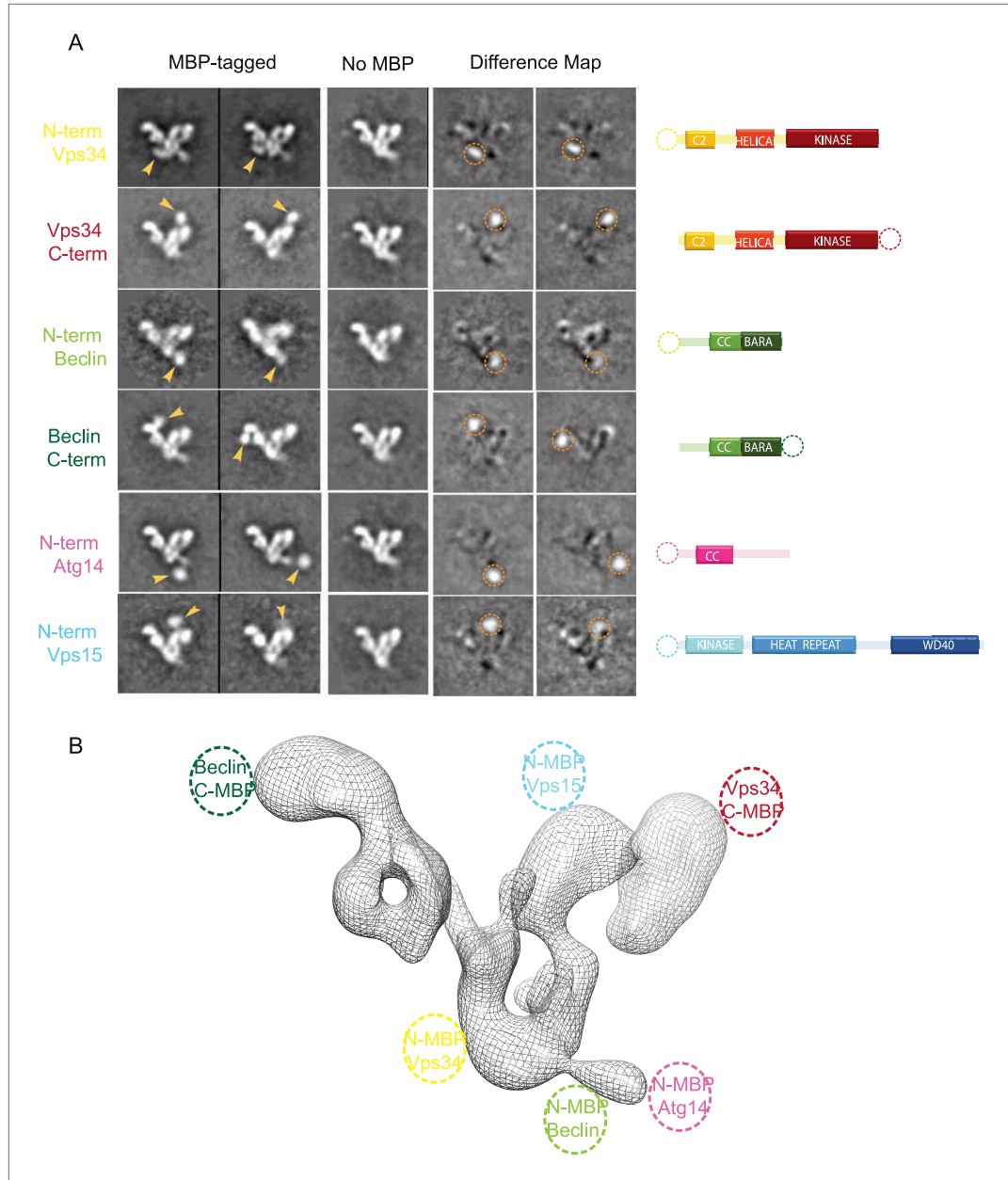

**Figure 4**. MBP-tagging identifies PI3KC3-C1 subunits. (**A**) Six different MBP-tagged versions of the complex were used to identify the position of the different PI3KC3-C1 subunits. A cartoon diagram indicating the position of the MBP tags is shown on the right. The two left columns show the reference-free 2D class averages of the MBP-labeled samples and arrowhead highlights the MBP tag position. The middle column shows the corresponding class for the unlabeled sample. Two right columns show the difference map calculated by subtracting the unlabeled reference class from the labeled and the dotted circle represents the MBP density. (**B**) 3D reconstruction of the PI3KC3-C1 complex highlighting the position of the six MBP tags used for domain mapping.

the crystal structure of the yeast Vps15 propeller domain (*Heenan et al., 2009*) (*Figures 4B and 5C*). Collectively, the docked or assigned regions of the VPS34 C2 and HELCAT, VPS15 kinase, HEAT, and WD40, BECN1 BARA domains account for 2162 residues of the 3187 present in ATG14–PI3KC3.

Two long, narrow tubes of density are present in the area where the ATG14 and BECN1 coiled coil regions would be expected (*Figure 5A,D* third panel, see arrows). One tube is adjacent to the VPS15 WD40 domain. It points towards the BECN1 BARA domain at one end and towards the second tube at the other. Following a gap in the density between the two tubes, the second one continues until it

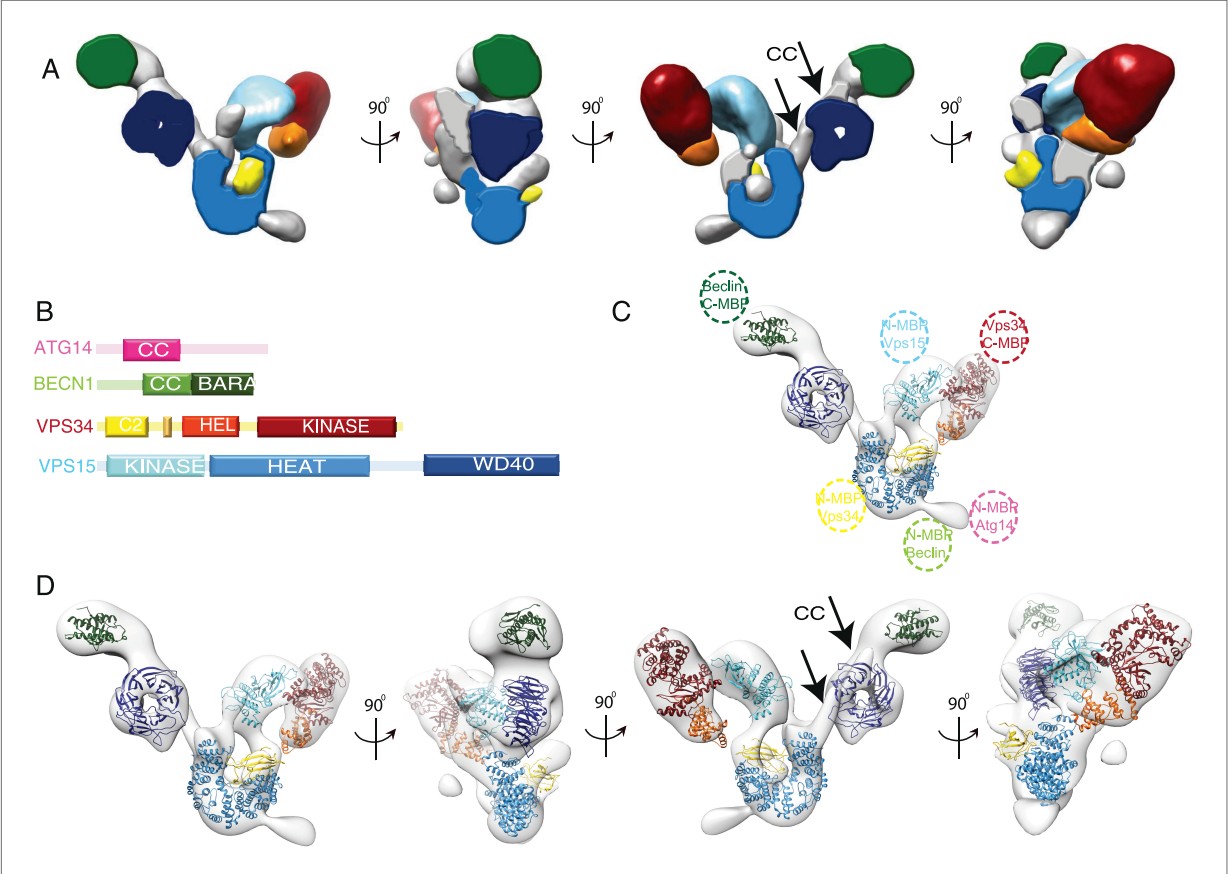

**Figure 5**. Subunit architecture of PI3KC3-C1. (**A**) Segmented volume representation of the 3D reconstruction highlighting the different domains of the complex colored as shown in (**B**). Arrows indicate regions assigned to the BECN1-ATG14 coiled coil. (**B**) Domain structures of the four subunits of PI3KC3-C1. (**C**) 3D reconstruction of the complex with the docked structures shown in a ribbon representation. The MBP label positions are represented as dotted circles. (**D**) 3D reconstruction of the complex with the docked structures in ribbon representation in different rotations.

merges with the density for the VPS15 HEAT domains. The BARA domain marks the C-terminus of the BECN1 coiled-coil, and MBP marks the N-termini of ATG14 and BECN1 coils, so we assigned this region to the coiled coil dimer. Because there is a break between the two tubular regions and the BARA density, we did not build an explicit molecular model of this region. One of the few unassigned density features projects away from the VPS15 HEAT domain (*Figure 5A,C*), very near the MBP markers for the N-termini of ATG14 and BECN1 (*Figure 4B*). This suggests that at least parts of the N-termini of one or both of these subunits have some ordered structure. The resulting model consists of a 190-Å long right arm and a 210-Å long left arm, joined at a 90° angle (*Figure 5* and *Video 1*).

## Conserved architecture of PI3KC3-C2

While both PI3KC3-C1 and -C2 have been implicated in autophagy, PI3KC3-C1 is the complex involved in autophagosome initiation (*Matsunaga et al., 2010*). UVRAG is larger than ATG14 and contains a unique N-terminal C2 domain (*Figure 6B*). In order to gain insight into the structural basis for the differences in localization and function between the two complexes, PI3KC3-C2 was expressed, purified, assayed, and prepared for EM imaging (*Figure 6A*) using the same procedures as for PI3KC3-C1. 2D class averages of this complex show the same characteristic V-morphology and preferential face-on orientations as seen for PI3KC3-C1. As for PI3KC3-C1, the majority of particles belonged to classes in which the VPS34 HELCAT domain was dislodged from the rest of the complex. The main difference in the images is the consistent presence of additional density at the base of the V (*Figure 6C*, arrows). The location of the density corresponds to the expected location of the UVRAG C2 domain. The main conclusion from the 2D characterization of the PI3KC3-C2 is that the overall conformation and architecture are essentially identical to that of the PI3KC3-C1 as seen at the present resolution of

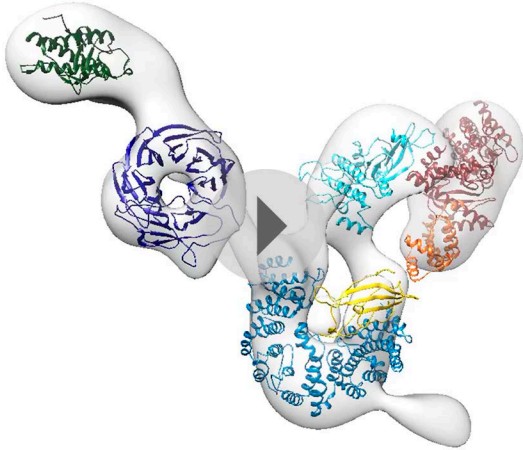

**Video 1**. Architecture of PI3KC3-C1. A 360° rotation of the docked model shown in **Figure 5D**.

our analysis. Therefore, the functional differences between the two complexes cannot be attributed to gross differences in architecture or conformation.

## Hydrogen exchange analysis of PI3KC3-C1 dynamics

Having assigned 65% of the density to known well-ordered regions of PI3KC3-C1 subunits, approximately 1025 out of the 3187 residues were not accounted for. These comprise the extended N-terminal regions of BECN1 and ATG14, the long C-terminal region of ATG14, and the linkers between the C2 and helical domains of VPS34 and the HEAT and WD40 domains of VPS15. In order to characterize these structurally undefined regions, we carried out hydrogen–deuterium exchange (HDX) as detected by mass spectrometry (MS) (**Englander, 2006**; **Engen, 2009**; **Chalmers et al., 2011**). Excellent peptide coverage was obtained for all of VPS15 (167 peptides were analyzed, 89.9% protein coverage) and all of VPS34 (119 peptides were analyzed, 74.7% protein coverage) with the exception of the C2 domain (**Figure 7A**). Throughout the ordered domains of VPS15 and VPS34, partial exchange was observed, consistent with the folded nature of the domains (**Figure 7A,B**). Complete exchange was observed throughout the VPS15 HEAT-WD40 linker at incubation times as short as 10 s (**Figure 7B**), indicating that this region is essentially completely flexible and intrinsically disordered. The linker between the helical and kinase domains of VPS34 also manifested complete exchange, consistent with the absence of density for this region in the VPS34 crystal structure (**Miller et al., 2010**).

One significant and unexpected zone of protection was discovered within the linker regions, a 'cold spot' comprising residues 218–237 of VPS34 (**Figure 1A**, yellow box between C2 and helical domains; **Figure 7A,C**). This region is part of the C2-helical domain linker. It is highly conserved from yeast to humans (**Figure 7D**), so we christened it the C2-Helical Internal Linker (CHIL) motif. Peptides covering this region show very little increase in deuteration over the interval from 10 s to 1 hr (**Figure 8**). Because this 20-residue motif is too small to form a folded entity on its own, we presume that it assembles with a nearby unit, likely the VPS15 HEAT or the VPS34 helical domain.

## Long range dynamics of VPS15 and VPS34

The 2D class averages calculated from all raw images of PI3KC3-C1 revealed a high degree of structural plasticity of the complex (**Figures 1D and 9**). Whilst a structurally homogeneous subset of images (**Figure 1D**, top row) was selected for the 3D reconstruction described above, the excluded images provide additional insights into the structural dynamics of PI3KC3-C1. Most strikingly, many class averages show the density identified as VPS34 HELCAT being dislodged from the rest of the complex (**Figure 1D**, lower two rows). Additionally, some class averages appear to entirely lack the density for VPS34 HELCAT. In principle, this could either be due to dissociation of VPS34 from the complex or due to the averaging out of HELCAT densities in a multitude of positions. The latter possibility seems more likely, given that the dislodged HELCAT positions generally lack contacts with the rest of the complex.

The untilted images recorded for the PI3KC3-C1 3D reconstruction were revisited to determine the abundance of states with a dislodged HELCAT. Of the particles that sorted into high-quality class averages, 48.5% had a lodged HELCAT, the remaining particles mainly having a dislodged but discernible HELCAT (**Figure 9D**). The same analysis of PI3KC3-C2 revealed 34.0% particles in the lodged state and a larger percentage of particles having no discernible HELCAT density (**Figure 9D**). The class averages in **Figure 1D** show the range of motion of the dislodged HELCAT, spanning a radius of up to 18 nm from the proposed position of the VPS34 C2 domain to the center of mass of the HELCAT. A second mode of flexibility of PI3KC3-C1 takes place at the VPS15 HEAT (**Figure 9**, **Video 2**). This flexibility is seen both in the lodged and dislodged HELCAT state and leads to a movement of the VPS15 kinase domain relative to the WD40. Comparison of individual class averages indicate that one

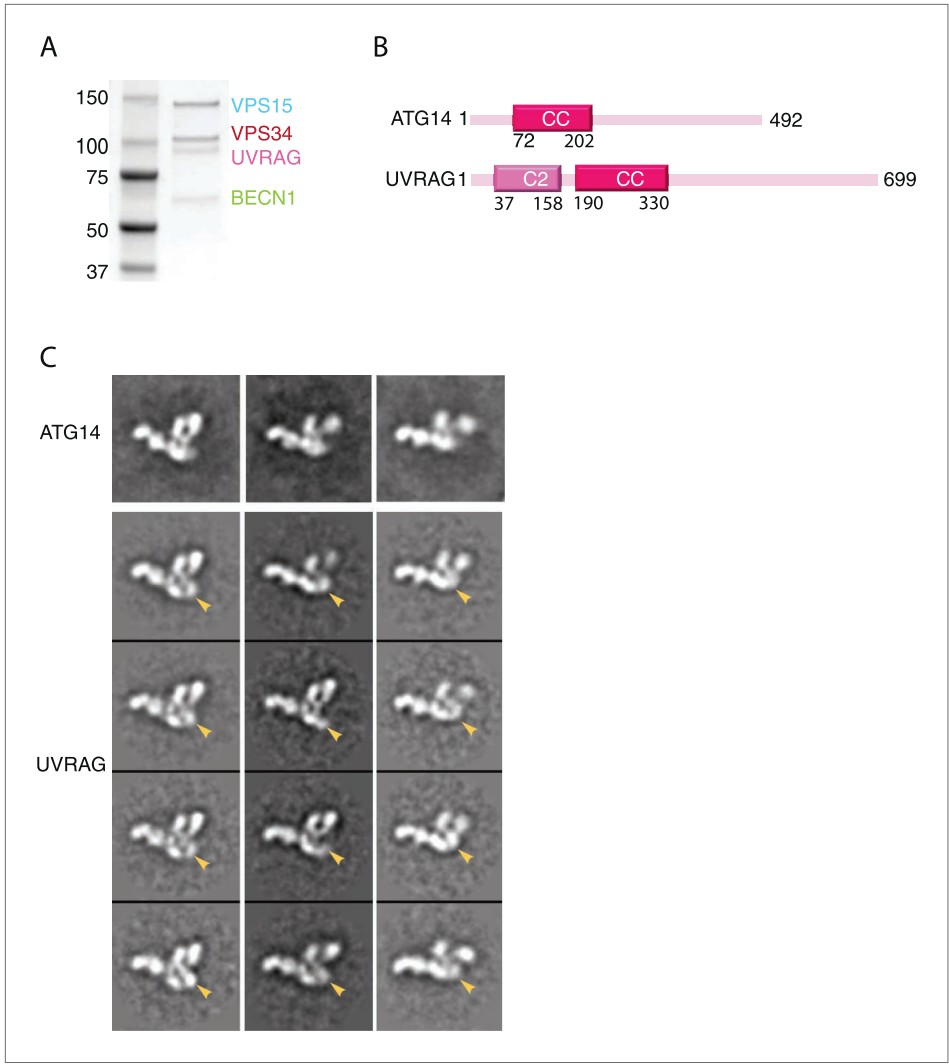

**Figure 6**. Reconstitution and EM analysis of PI3KC3-C2. (**A**) Purification of PI3KC3-C2. Coomassie-stained SDS-PAGE gel of purified PI3KC3-C2. (**B**) Domain structures of ATG14 and UVRAG. (**C**) The top row shows the reference-free 2D class averages for the PI3KC3-C2 complex. The bottom four rows show the corresponding class averages of the PI3KC3-C2 and arrowhead highlights the additional density attributed to the UVRAG C2 domain.

pivoting point for this motion is between the WD40 domain and the C-terminus of the HEAT repeat, with additional flexibility between the HEAT and kinase domains (*Video 2*).

## Discussion

The EM reconstruction and docking model for PI3KC3-C1 provides the first overall view of VPS15, VPS34, BECN1, and ATG14 in their functional context. The HDX-MS experiments together with the EM analysis highlight the dynamic nature of the complex. At the available resolution, the overall architecture is essentially the same for PI3KC3-C2 and PI3KC3-C1. The reconstruction thus provides a unified model for VPS34 complex organization. The overall structure allows us to rationalize a variety of previous data on subunit organization. The BARA domain of BECN1 is essential for the incorporation of BECN1 into the PI3KC3-C1 complex (*Furuya et al., 2005*), consistent with its interactions with the VPS15 WD40 domain in the EM structure. Similarly, the ATG14 coiled coil region is essential for its association with VPS34 (*Itakura et al., 2008*), consistent with the interactions of both the ATG14 coiled coil and VPS34 with the central VPS15 scaffold. The C2 domain of VPS34 was previously shown to be the main locus for the incorporation of VPS34 into the complex (*Liang et al., 2006*). This fits with

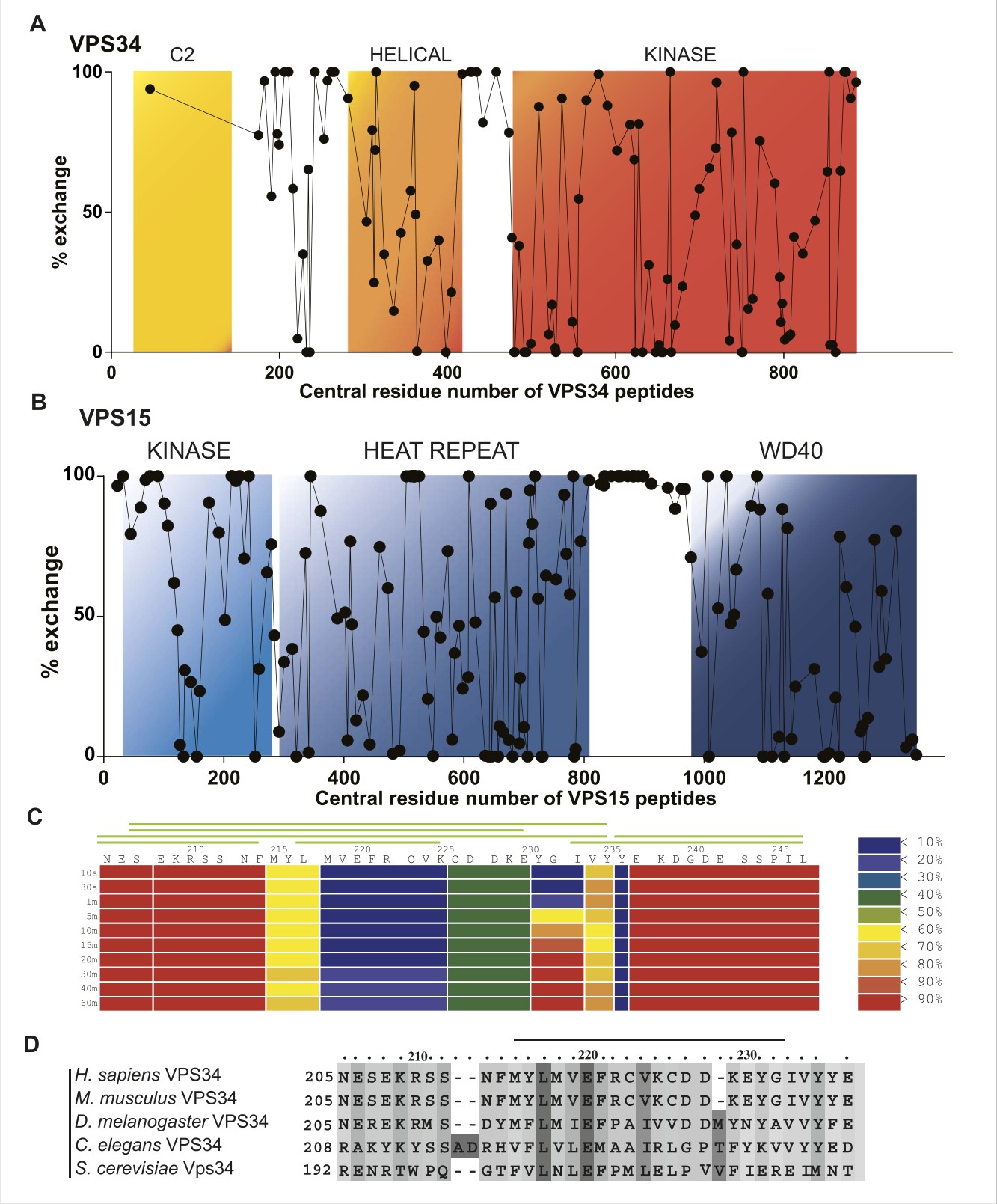

**Figure 7**. Local dynamics of PI3KC3-C1 assessed by hydrogen–deuterium exchange. PI3KC3-C1 was incubated for 10 s in $D_2O$. Percent deuterium incorporation plotted vs the central residue number of peptides from VPS34 (**A**) and VPS15 (**B**). (**C**) Percent change in deuteration for VPS34 region 205–247 across various time points (10 s to 1 hr). Green bars above refer to peptide coverage. (**D**) Conservation of the C2-Helix Internal Linker (CHIL) motif in VPS34.

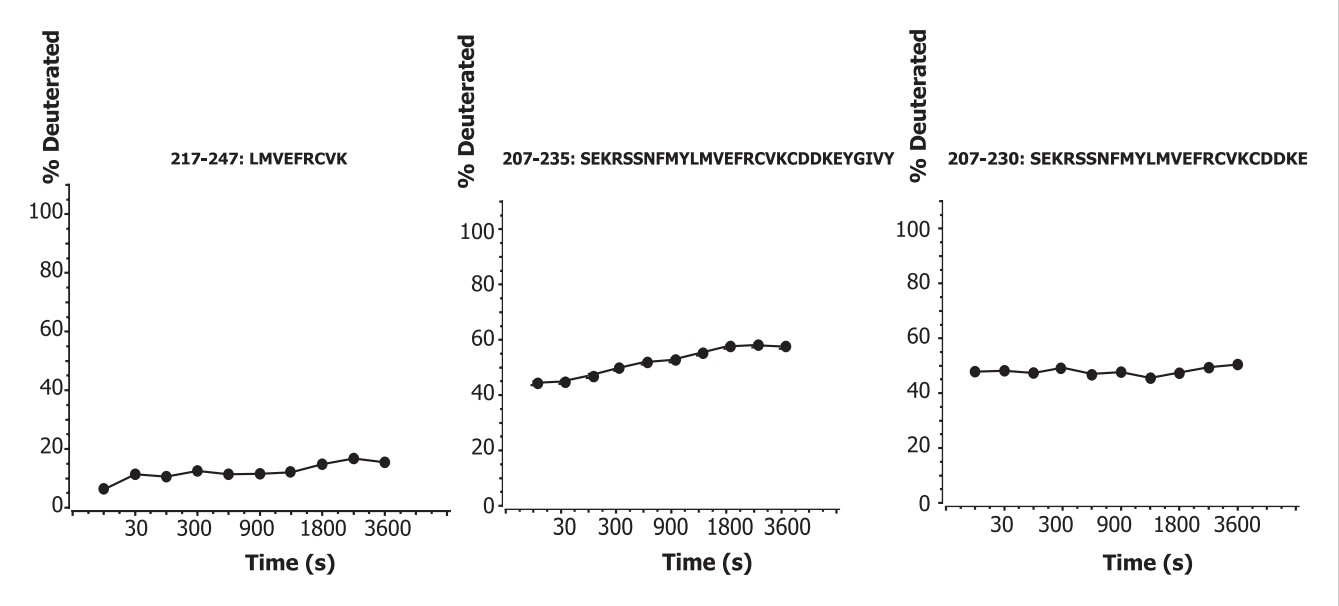

**Figure 8**. Time course of deuteration of CHIL motif peptides. Hydrogen–deuterium incorporation as a function of time (10 s to 1 hr) for selected peptides in the CHIL motif of VPS34. The deuteration level of an individual peptide was determined via HDexaminer, and it was assumed that the first residue in a peptide is non-deuterated.

the structural observation that VPS34-C2 is integral to the core architecture, while the C-terminal HELCAT region is readily dislodged. An interaction was reported between the C-terminal helix of VPS34 and the VPS15 kinase-HEAT region (**Budovskaya et al., 2002**), consistent with the contacts we observed between the kinase domains of VPS15 and VPS34. The C-terminal helix of VPS34 is critically involved in membrane interaction upon activation (**Miller et al., 2010**), and this raises the interesting speculation that VPS34 could be regulated via inter-subunit sequestration of the same helix. The congruence of the size and shape of density features with subunit domains, the connectivity of the domains within VPS34 and VPS15, the localization of the tags, and the consistency with published interactions all lend high confidence to our structural assignments and the defined architecture of the complex.

The 2D class averages highlight the globally dynamic nature of the VPS34 HELCAT module relative to the rest of the complex. Many of these class averages show that VPS34-HELCAT is dislodged from the rest of the complex, even though the rest of the complex is intact. The class averages also provide some insight into the nature of the motions. It should be emphasized that, while there is evidence for domain movements, the exact extent and direction of the movements are unknown. The range of movements may be constrained in our experiments by the presence of the carbon film on which the complex is imaged. Thus, the movements noted here may be a subset of the range that could occur in solution. VPS34-HELCAT makes contacts principally with the kinase domain of VPS15. In many of the class averages, the VPS15 kinase domain pivots toward the other arm of the V. The pivoting motion includes the VPS15 HEAT repeat domain. Thus the VPS15 kinase and HEAT regions function as a single module. By analogy to the HELCAT terminology for VPS34, we refer to it as the KINHEAT module. VPS15 is not obviously a pseudokinase on the basis of its sequence (**Taylor et al., 2013**), yet there is no compelling evidence that VPS15 can phosphorylate substrates (**Backer, 2008**). The VPS15-KINHEAT physically contacts the VPS34-HELCAT region, which suggests that regulation could occur through direct and stable interactions without covalent phosphorylation.

While the V-shaped PI3KC3-C1 complex flexes on a large scale, the EM and HDX data also show that there are limits to the types of motions that occur. The HDX data show that the VPS15 HEAT-WD40 linker is essentially completely disordered, yet the WD40 domain is firmly lodged in the complex in all class averages. The WD40 is thus clearly not undergoing tethered diffusion or 'fly-casting' in the context of the assembled complex. The only part of the structure that we observed to undergo fly-casting is the VPS34 HELCAT module. Even in this case, the motions are constrained. Although the linker between the VPS34 C2 and helical domains spans 130 residues, the presence of the anchored CHIL

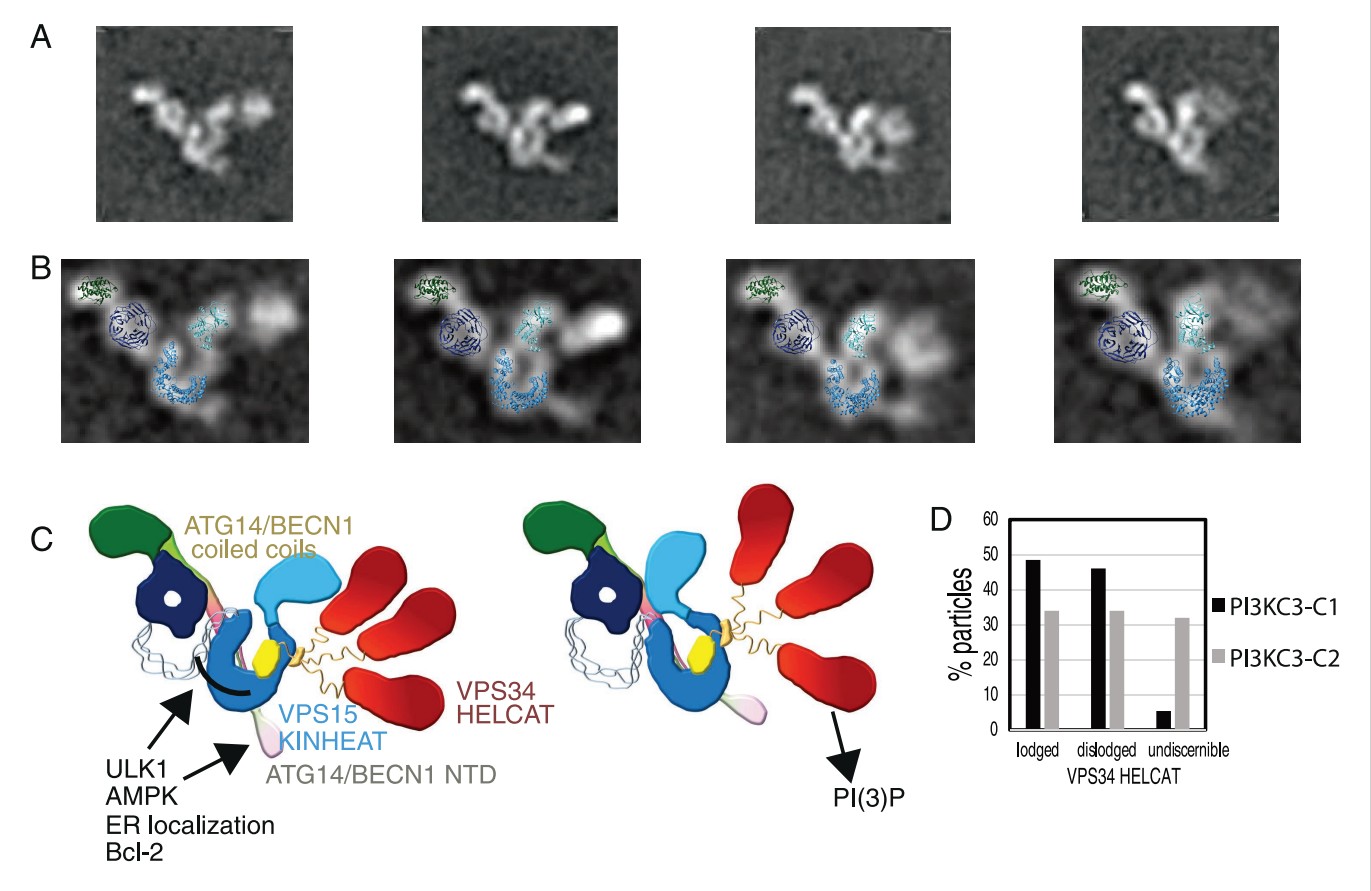

**Figure 9**. Global conformational changes of PI3KC3-C1. (**A**) Representative class averages of the PI3KC3-C1 complex with the VPS15 KINHEAT in the open (far left) to closed (far right) conformation and the dynamic VPS35 HELCAT. (**B**) Class averages with docked VPS15 KINHEAT, VPS15 WD-40, and BECN1 BARA domains overlaid. (**C**) Schematic of the PI3KC3-C1 complex showing the open and closed conformations of the Vps15 KINHEAT and the dynamics of the Vps34 HELCAT. The VPS34 C2 domain is yellow and the CHIL motif is orange. (**D**) Percentage of well-resolved particles sorting into class averages with a lodged, dislodged, or undiscernible VPS34 HELCAT.

domain in the center of this linker sharply restricts the range of motion. There are ~50–65 residues between the CHIL and the boundaries of the nearest ordered modules, which would limit the maximum reach of the fully extended linker to ~20 nm. This is consistent with the maximum 18 nm displacement of the HELCAT region seen in EM images.

It is striking that no direct contacts are seen between the ATG14 and BECN1 module and VPS34. Given the resolution of the reconstruction, it is certainly possible that direct interactions exist though they were not visualized. Nevertheless, the structure highlights a central and extensive role for VPS15 in scaffolding and bridging between the ATG14:BECN1 regulatory subcomplex and the VPS34 catalytic subunit. VPS15 is capable of pivoting about the KINHEAT-WD40 junction. The base of the V is mainly formed from the HEAT repeats of VPS15. The base of the V is also the locus for the unstructured or unresolved N-terminal regions of ATG14 and BECN1.

Most of the known signaling inputs into PI3KC3-C1 converge on the N-terminus of BECN1. This region is subject to activating phosphorylations by the upstream autophagy initiating kinase ULK1 (*Russell et al., 2013*) and the AMP-activated kinase AMPK (*Kim et al., 2013*). The BECN1 BH3 domain responsible for binding to Bcl-2, which inhibits autophagy, is located in this region (*Pattingre et al., 2005*). Along the same lines, a Cys-rich region of ATG14 immediately N-terminal to the coiled coil has been proposed to be the site of ER targeting (*Matsunaga et al., 2010*). Thus, this region of the PI3KC3-C1 structure might act as an allosteric reporter for localization to the ER membrane. It is tempting to speculate that the right arm of the complex transmits signals from the region of the BECN1 N-terminus to the VPS34 lipid kinase domain. It is exciting to finally have a structural context

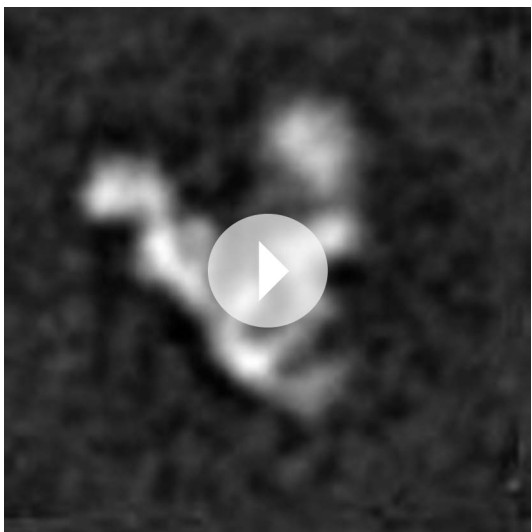

**Video 2**. Conformational changes of PI3KC3-C1. A sequence of 2D class averages arranged according to the pivoting motion of VPS15, from open to more-closed. The sequence also shows several different lodged and dislodged states of VPS34 HELCAT (in no particular order).

in which to test how the many signals that impinge on BECN1 and PI3KC3-C1 are processed.

## Materials and methods

### Protein expression and purification

The full length DNAs encoding VPS15, VPS34, BECN1, ATG14, and UVRAG were codon optimized for expression in HEK293 cells. Synthetic genes encoding VPS15, VPS34, BECN1 DNAs were amplified by PCR and cloned into the pCAG vector coding for an N-terminal twin-STREP-FLAG tag using KpnI and XhoI restriction sites. The ATG14 and UVRAG DNAs were cloned in the pLEXm vector encoding an N-terminal GST tag followed by TEV restriction site. The pCAG vector encoding an N-terminal GST tag followed by a TEV restriction site was used for expression of ATG14. HEK293 cells adapted for suspension were grown in Freestyle media (Invitrogen, Grand Island, NY) supplemented with 1% FBS (Invitrogen) at 37°C, 80% humidity, 5% $CO_2$, and rocked at 140 rpm. Once the cultures reached 1.5–2 million cells ml$^{-1}$ in the desired volume, they were transfected as followed. For a 1 l transfection, 3 ml PEI (1 mg ml$^{-1}$, pH 7.0) was added to 33 ml Hybridoma media and 1 mg of total DNA in another 33 ml hybridoma media. 1 mg of transfection DNA contained equal mass ratio of PI3KC3-C1 and -C2 complex expression plasmids (pLEXm-GST-TEV-ATG14 [or pLEXm-GST-TEV-UVRAG], pCAG-twinSTREP-FLAG-VPS15, pCAG-twinSTREP-FLAG-VPS34, and pCAG-twinSTREP-FLAG-BECN1). PEI was added to the DNA, mixed and incubated for a further 20 min at room temperature. 66 ml of the transfection mix was then added to each 1 l culture. Cells were harvested after 3 days.

Cells were lysed by gentle shaking in lysis buffer (50 mM Tris, pH 8.0, 200 mM NaCl, 2 mM MgCl$_2$, 10% (vol/vol) glycerol, 1% (vol/vol) Triton X-100, 1 mM TCEP, and EDTA free proteinase inhibitors [Roche, Basel, Switzerland]) at 4°C. Lysates were clarified by centrifugation (18,000×$g$ for 60 min at 4°C) and incubated with 10 ml glutathione Sepharose 4B (GE Healthcare, Uppsala, Sweden) for 1 hr at 4°C with gentle shaking. The glutathione Sepharose 4B matrix was applied to a gravity column, washed four times with 50 ml wash buffer (50 mM Tris, pH 8.0, 200 mM NaCl, 2 mM MgCl$_2$, and 1 mM TCEP), and purified complexes were eluted with 50 ml wash buffer containing 50 mM reduced glutathione. Eluted complexes were treated with TEV protease at 4°C overnight. TEV-treated complexes were loaded on a 2.5 ml Strep-Tactin Sepharose gravity flow column (IBA GmbH, Göttingen, Germany; at 4°C). The Strep-Tactin Sepharose column was washed five times with 2.5 ml wash buffer, and purified complexes were eluted with 6 ml wash buffer containing 10 mM desthiobiotin (Sigma-Aldrich, St. Louis, MO). Eluted complexes were purified to homogeneity by injection on Superose 6 16/50 (GE Healthcare) column that was pre-equilibrated in gel filtration buffer (20 mM Tris–HCl, pH 8.0, 200 mM NaCl, 2 mM MgCl$_2$, and 1 mM TCEP).

### Lipid kinase assay

ATP consumption in the presence of lipids was determined using the ADP-Glo Kinase Assay kit (Promega, Madison, WI). The kinase reaction was performed in 96-well NBS white plates (Corning, Corning, NY), in 1.25× kinase reaction buffer (12.5 mM HEPES, pH 7.0, 125 mM NaCl, 2.5 mM MnCl$_2$), and 2 µg PI:phosphatidylserine (PS) substrate solution. The reaction was initiated by adding 5 µl of 125 µM ATP to 20 µl of 1.25× kinase reaction buffer. The reaction was carried out for 1 hr at 37°C, then 25 µl of ADP-Glo reagent containing 10 mM MgCl$_2$ was added to the reaction mixture and incubated at 23°C for 30 min to stop the enzyme reaction and deplete unconsumed ATP. After depletion of ATP, 50 µl of Kinase Detection Reagent was added to convert ADP to ATP and introduce luciferase and luciferin to detect ATP. The reaction mixture was further incubated 23°C for 30 min and the luminescence was measured with a GloMax-Multi detection system (Promega).

Production of PI(3)P from PI was assayed as follows. Recombinant PI3KC3 was pre-incubated in 73 µl reaction buffer (20 mM Tris–HCl, pH 8.0, 100 mM NaCl, 10 mM $MgCl_2$) containing 20 µg sonicated phosphatidylinositol (Avanti Polar Lipids, Alabaster, AL) for 20 min on ice. The reaction was started by adding 6 µl cold ATP (0.5 mM in reaction buffer) and 1 µl ATP [γ-32P] (10 µCi, PerkinElmer). After incubation at room temperature for 20 min, the reaction was terminated by the adding 20 µl 8 M HCl. The organic phase was extracted with 160 µl methanol/chloroform (1:1, vol/vol). Extracted phospholipid products were resolved by TLC using a silica-coated gel (Whatman, Little Chalfont, United Kingdom) and a solvent composed of chloroform:methanol:4 M ammonium hydroxide (vol/vol/vol, 9/7/2), followed by visualization with a phosphorimager (Typhoon Trio, GE Healthcare).

## Electron microscopy sample preparation

Negatively stained samples of PI3KC3-C1 and -C2 were prepared on continuous carbon grids that had been plasma cleaned in a 10% $O_2$ atmosphere for 10 s using a Solarus plasma cleaner (Gatan Inc., Pleasanton, CA). 4 µl of PI3KC3 complexes at a concentration of 25 nM in 20 mM Tris, pH 8.0, 200 mM NaCl, 2 mM $MgCl_2$, 1 mM TCEP, and 3% trehalose were placed on the grids and incubated for 30 s. The grids were floated on four successive 50 µl drops of 1% uranyl formate solution incubating for 10 s on each drop. The stained grids were blotted to near dryness with a filter paper and air-dried.

## EM data collection

For the MBP tag analysis, native and tagged samples were imaged using an FEI Tecnai 12 electron microscope (FEI, Hillsboro, OR) operated at 120 keV at a nominal magnification of 49,000 (2.18 Å calibrated pixel size at the specimen level) using a defocus range of −0.7 to −1.5 µm with an electron dose of $35e^-/Å^2$. Images were acquired on a TVIPS TemCam F-416 4049 × 4096 pixel CMOS detector (TVIPS GmbH, Gauting, Germany) using the automated Leginon data collection software (*Suloway et al., 2005*). For the initial 3D random conical tilt reconstruction, tilt-pair images were recorded at 0° and 50° at 30,000 nominal magnifications with the same setup (3.56 Å/pixel at the specimen level). Subsequent 3D structure refinement was performed using larger datasets with images acquired using an FEI Tecnai F20 transmission electron microscope operated at 120 kV at a nominal magnification of 80,000 using a defocus range of −0.7 to −1.5 µm with an electron dose of $35e^-/Å^2$ at tilts of 0°, 30°, and 45°. Images were acquired on a Gatan UltraScan4000 4049 × 4096 pixel CCD detector using Leginon with a 1.51 Å calibrated pixel size at the specimen level (*Suloway et al., 2005*).

## Image processing

The initial steps of image processing and classification were performed using the Appion image processing environment (*Lander et al., 2009*). Particles were first selected manually from micrographs and subjected to 2D iterative reference-free classification and alignment using a topology-representing network classification and IMAGIC (Image Science Software GmbH, Berlin, Germany) multi-reference alignment (MRA) (*van Heel et al., 1996*; *Ogura et al., 2003*). The 2D class averages thus generated served as templates for subsequent automated particle selection. Template-based automated particle picking was performed using the FindEM program (*Roseman, 2004*). The contrast transfer functions (CTFs) of the micrographs were estimated using the CTFFIND for untilted images and CTFTILT for tilted images (*Mindell and Grigorieff, 2003*). CTF correction of the micrographs was performed by Wiener filter using ACE2 (*Mallick et al., 2005*) for the micrographs used for subunit localization and by tilt- and position-dependent phase flipping using EMAN (*Ludtke et al., 1999*) for micrographs used for the 3D reconstructions. Particles were extracted using a 192 × 192 (T12) or 256 × 256 (F20) pixel box size and binned by a factor of 2. Each particle was normalized to remove pixels whose values were above or below 4.5 σ of the mean pixel value using the XMIPP normalization program (*Scheres et al., 2008*). In order to remove incorrectly selected protein aggregates or other artifacts, images whose mean or standard deviation deviated a lot from the typical values of the dataset were removed. The remaining particles were subjected to five rounds of iterative classification using a topology-representing network and 2D MRA in IMAGIC (*van Heel et al., 1996*; *Ogura et al., 2003*). The resulting 2D class averages were manually inspected to remove protein aggregates, contaminants, lower-resolution classes, as well as particles with a dislodged VPS34 HELCAT domain. The remaining particles were subjected to another five rounds of classification and MRA to produce the final 2D class averages, each being calculated from an average of 200–250 raw images.

## EM analysis of tagged complexes

Datasets consisting of 53,649, 36,997, 3799, 20,065, 40,854, and 12,804 particles were collected for the MBP-VPS34, VPS34-MBP, MBP-BECN1, BECN1-MBP, MBP-ATG14, and MBP-VPS15, respectively. For the subunit and domain localization, reference-free 2D class averages were generated as described above for each tagged dataset and the native dataset. Class averages from each dataset were grouped by projection matching to re-projections of a filtered reference volume of the native complex using SPIDER (*Shaikh et al., 2008*). The corresponding class averages in the different projection groups of the native and tagged datasets were manually examined to locate the position of the tag. For the difference map generation class averages in the same projection group were individually normalized and the native class averages were subtracted from the tagged class averages. Extra density for the MBP tags was clearly visible in 31% of the class averages for MBP-VPS34, 9.1% for VPS34-MBP, 10% for MBP-BECN1, 23.5% for BECN1-MBP, 14% for MBP-ATG14, and 3.6% for MBP-VPS15.

## Three-dimensional structure determination

An ab initio structure was determined by random conical tilt (RCT) using SPIDER routines integrated into Appion (*Radermacher et al., 1987*; *Lander et al., 2009*). Briefly, 23 tilt-pair images yielded an initial dataset of 12,027 tilt-pair particles. The untilted particles were aligned and classified in 2D using reference-free classification and MRA as described above, followed by a single round of 2D reference-based alignment using Spider's AP MQ command (*Frank et al., 1996*). The final 14 classes ranged from 725 to 1198 untilted particles each and class-volumes were calculated from the matching tilted particles by backprojection. The class-volumes were then refined against the entire dataset (both tilted and untilted) using an iterative projection matching procedure combining SPARX and EMAN2 libraries (*Baldwin and Penczek, 2007*; *Tang et al., 2007*). Each class-volume that corresponded to the whole complex converged to the same 3D reconstruction (with a final resolution of 27–30 Å, according to the FSC at 0.5 cutoff criterion). This map was used as the starting model for all subsequent structure refinements. Structure refinement was performed using images acquired at 0°, 30°, and 45°. The initial particle selection gave 109,776 particles at 0°, 24,085 at 30°, and 25,694 at 45°. Reference-free 2D class averages were calculated as described above, with images recorded at different tilt angles treated separately to allow assessment of tilt-dependent orientations and image quality. Initial attempts to perform 3D classification using RELION (*Scheres, 2012*) with respect to the position on the VPS34 HELCAT region did not result in robust 3D classes. Thus, particles belonging to class averages of lower resolution, as well as class averages with a dislodged VPS34 HELCAT region, were excluded from further processing. This resulted in a dataset of 24,116 particles at 0°, 8643 at 30°, and 5986 at 45°. The final structure (deposited at EMDB with accession number 2846) was calculated using a single 3D class in RELION (*Scheres, 2012*), with the main RCT structure low-pass filtered to 80 Å as initial reference. The distribution of Euler angles assigned by RELION was visualized with UCSF Chimera (*Pettersen et al., 2004*). The structure had a resolution of 28 Å according to the gold standard FSC criterion as implemented in RELION and was used for docking of homology models of domains. The density map was further validated by tilt-pair analysis (*Rosenthal and Henderson, 2003*) with image pairs collected at 0° and 15° using software made available by John Rubinstein (U Toronto). As a control for the tilt-pair analysis, image pairs were collected on negatively stained *Escherichia coli* 70S ribosomes under identical conditions, co-mapping them to a published 70S ribosome map (EMDB 1849) (*Agirrezabala et al., 2011*).

## HDX-MS

Amide hydrogen exchange mass spectrometry (HDX-MS) was initiated by a 20-fold dilution of stock PI3KC3-C1 (2 μM) into $D_2O$ buffer containing 20 mM Tris–HCl (pD 8.0), 200 mM NaCl, 2 mM $MgCl_2$, and 1 mM TCEP at 30°C. Incubations in deuterated buffer were performed at intervals from 10 s to 1 hr. Backbone amide exchange was quenched at 0°C by the addition of ice-cold quench buffer (400 mM $KH_2PO_4/H_3PO_4$, pH 2.2). Quenched samples were injected onto a chilled HPLC setup with in-line peptic digestion and desalting steps. Desalted peptides were eluted and directly analyzed by an Orbitrap Discovery mass spectrometer (Thermo Scientific, Waltham, MA). The HPLC system was extensively cleaned between samples. Initial peptide identification was performed via tandem MS/MS experiments. A PEAKS Studio 7 (www.bioinfor.com) search was used for peptide identification. Initial mass analysis of the peptide centroids was performed using HDExaminer version 1.3 (Sierra Analytics, Modesto, CA), followed by manual verification of each peptide. The deuteron content of the peptic peptides covering PI3KC3-C1 was determined from the centroid of the molecular ion isotope envelope. The deuteron

content was adjusted for deuteron gain/loss during pepsin digestion and HPLC. Both non-deuterated and fully deuterated PI3KC3-C1 were analyzed. Fully deuterated PI3KC3-C1 was prepared by four cycles of drying and resolubilization in $D_2O$ containing 6 M guanidinium hydrochloride.

## Acknowledgements

This work was supported by National Institutes of Health grant GM111730 (JHH). RES was supported by a Damon Runyon Cancer Research Fellowship and a L'Oreal USA Women in Science Fellowship. L-AC was supported by a Long-Term Fellowship from the Human Frontiers Science Program (LT001037/2011-L). We thank David Taylor for assistance with the tilt-pair validation. EN is a Howard Hughes Medical Institute Investigator.

## Additional information

### Funding

| Funder | Grant reference number | Author |
|---|---|---|
| National Institute of General Medical Sciences | GM111730 | James H Hurley |
| Howard Hughes Medical Institute (HHMI) | | Eva Nogales |
| Damon Runyon Cancer Research Foundation | Fellowship | Robin E Stanley |
| L'Oreal USA | Women in Science Fellowship | Robin E Stanley |
| Human Frontier Science Program | Long-Term Fellowship (LT001037/2011-L) | Lars-Anders Carlson |

The funders had no role in study design, data collection and interpretation, or the decision to submit the work for publication.

### Author contributions

SB, L-AC, GS, Conception and design, Acquisition of data, Analysis and interpretation of data, Drafting or revising the article, Contributed unpublished essential data or reagents; LNY, DJK, Acquisition of data, Analysis and interpretation of data, Contributed unpublished essential data or reagents; PG, Acquisition of data, Analysis and interpretation of data; RES, Conception and design, Contributed unpublished essential data or reagents; EN, JHH, Conception and design, Analysis and interpretation of data, Drafting or revising the article

### Author ORCIDs

Goran Stjepanovic, ⓘD http://orcid.org/0000-0002-4841-9949

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
