## [Decision Letter]

Thank you for sending your work entitled “Architecture and Dynamics of the Autophagic Phosphatidylinositol 3-Kinase Complex” for consideration at *eLife*. Your article has been favorably evaluated by Tony Hunter (Senior Editor), a Reviewing editor, and 2 reviewers.

The Reviewing Editor and the reviewers discussed their comments before we reached this decision, and the Reviewing Editor has assembled the following comments to help you prepare a revised submission.

The study presents a significant step forward towards a structural understanding of the autophagy process. Despite the fact that the structure is of low resolution and mechanistic insights are limited, it is of significant interest as it is technically challenging and many difficulties in complex preparation and image analysis had to be overcome.

Substantive concerns:

1) The inherent limitations of the EM data quality are not discussed. These limitations should be made clear to a wider audience, as the current presentation may be misleading for non-specialists. For example, the map (Figure 1) appears to be distorted in one direction, which is likely the combined result of preferential orientation, missing views, and flattening in negative stain. In addition, the missing cone makes the map resolution anisotropic. As a result, the WD40 domain envelope is much wider than expected from the fit. A more thorough discussion of resolution anisotropy is needed.

2) The stated resolution of 26 Å appears to be an overestimate, perhaps again as a result of anisotropy. Although the WD40 propeller is resolved, the resolution of other map regions is likely to be significantly lower, given the dynamic nature of the complex. Currently, the FSC up to 1/40 Å is about 1 and drops abruptly to 0.143 within 5 Fourier pixels. This RELION-specific is hardly appropriate for low resolution maps in negative stain. The FSC should be recalculated by adjusting the low-resolution cutoff to lower spatial frequencies for a more realistic falloff in resolution. Alternatively, a more conservative mask may provide a more realistic resolution estimate.

3) The presentation of the model, which has several major gaps and uncertainties, is overconfident (e.g. “it was possible to account for almost all of the EM density” or “Having assigned nearly all of the electron density features to known well-ordered regions”). Furthermore, in Figure 5 parts of the model are color-coded, for which there are no atomic models (cf Figure 5). Figure 5 should be revised to leave unassigned densities white. As the exact orientations and connections of the domains are uncertain, the most appropriate presentation of the model is Figure 5, which highlights the tags that have been used to locate the domains. The legend to Figure 5 should state that at the current resolution, the fit of the x-ray structures can be only approximate. Moreover, the authors should quantify the percentage of the EM volume that is reliably accounted for by atomic models.

4) Some of the apparent dynamic behaviour of the complex may simply be the result of carbon film/specimen interactions in negative stain. For example, the images show that the movement of the Vps34 HELCAT domain is limited to in-plane rotation parallel to the carbon film, but Figure 9 presents a schematic model that suggests a functional role of this particular movement. It should be made clear that, while there is evidence for domain movements, the exact extent and direction of the movement is unknown.

5) Only about 50% of the complexes form a V-shape. In other class averages, the HEAT-repeat and kinase domain portion of Vps34 is dislodged/separated from the rest of the complex and VPS15 pivots at its position at the base of the V. The authors interpret these changes either as the complex falling apart (this happens to lots of complexes on an EM grid) or else, and this model is very extensively developed, flexibility indicating allosteric communication via the VPS15 HEAT repeat domain. There is, however, no data that supports the latter possibility over the former at this stage.

6) The highly speculative discussion of allostery should be shortened. As the evidence for allostery is very weak, the authors should remove or at least tone down the last sentence of the abstract (“This suggests a mechanism for the allosteric communication via the kinase and HEAT domains of VPS15”).

---

## [Author Response]

*1) The inherent limitations of the EM data quality are not discussed. These limitations should be made clear to a wider audience, as the current presentation may be misleading for non-specialists. For example, the map (*Figure 1*) appears to be distorted in one direction, which is likely the combined result of preferential orientation, missing views, and flattening in negative stain. In addition, the missing cone makes the map resolution anisotropic. As a result, the WD40 domain envelope is much wider than expected from the fit. A more thorough discussion of resolution anisotropy is needed*.

The issue of anisotropy and the missing cone is now addressed in a new sentence in the last paragraph of the “reconstitution and imaging of PI3KC3-C1” section.

*2) The stated resolution of 26 Å appears to be an overestimate, perhaps again as a result of anisotropy. Although the WD40 propeller is resolved, the resolution of other map regions is likely to be significantly lower, given the dynamic nature of the complex. Currently, the FSC up to 1/40 Å is about 1 and drops abruptly to 0.143 within 5 Fourier pixels. This RELION-specific is hardly appropriate for low resolution maps in negative stain. The FSC should be recalculated by adjusting the low-resolution cutoff to lower spatial frequencies for a more realistic falloff in resolution. Alternatively, a more conservative mask may provide a more realistic resolution estimate*.

We agree that the default of 1/40Å as a de facto starting point for the gold standard FSC calculation was not ideal in this case. We recalculated the FSC, averaging half-maps up to 1/60Å and updated Figure 3 accordingly. While the new FSC curve looks very similar, the recalculation changed the nominal resolution to 27.5 Å, and we have updated the text accordingly. As for the mask, we would like to note that the only mask used during the structure calculation was a spherical mask with a soft edge as implemented in RELION and a radius of 166 Å. This mask will not artificially boost the resolution calculation.

*3) The presentation of the model, which has several major gaps and uncertainties, is overconfident (e.g. “it was possible to account for almost all of the EM density” or “Having assigned nearly all of the electron density features to known well-ordered regions”). Furthermore, in*
Figure 5
*parts of the model are color-coded, for which there are no atomic models (cf*
Figure 5*).*
Figure 5
*should be revised to leave unassigned densities white. As the exact orientations and connections of the domains are uncertain, the most appropriate presentation of the model is*
Figure 5*, which highlights the tags that have been used to locate the domains. The legend to*
Figure 5
*should state that at the current resolution, the fit of the x-ray structures can be only approximate. Moreover, the authors should quantify the percentage of the EM volume that is reliably accounted for by atomic models*.

We calculated the percentage of the volume accounted by the atomic structure as follows. A density map was generated for the integrated docked PDB structure model and the volume of the density was computed in chimera, with the threshold of the volume manually set so that the surface coincided with the docked PDB structure model. The calculated density accounted for 65% of the actual experimental density volume. The text mentioned has been modified. Figure 5 has been redone to leave the unaccounted density regions in grey as suggested by the reviewers. The region assigned to the coiled coil is now colored white, since it was not fitted with an atomic model. We have retained the label “CC”, however, since we are confident of the assignment.

*4) Some of the apparent dynamic behaviour of the complex may simply be the result of carbon film/specimen interactions in negative stain. For example, the images show that the movement of the Vps34 HELCAT domain is limited to in-plane rotation parallel to the carbon film, but*
Figure 9
*presents a schematic model that suggests a functional role of this particular movement. It should be made clear that, while there is evidence for domain movements, the exact extent and direction of the movement is unknown*.

These are excellent points and we have taken the liberty of paraphrasing some of the wording above in the second paragraph of the discussion.

*5) Only about 50% of the complexes form a V-shape. In other class averages, the HEAT-repeat and kinase domain portion of Vps34 is dislodged/separated from the rest of the complex and VPS15 pivots at its position at the base of the V. The authors interpret these changes either as the complex falling apart (this happens to lots of complexes on an EM grid) or else, and this model is very extensively developed, flexibility indicating allosteric communication via the VPS15 HEAT repeat domain. There is, however, no data that supports the latter possibility over the former at this stage*.

As described below under point 6, we have sharply reduced the amount of speculation on the possible role of VPS15 in transmitting signals. What remains is explicitly couched as speculation.

*6) The highly speculative discussion of allostery should be shortened. As the evidence for allostery is very weak, the authors should remove or at least tone down the last sentence of the abstract (“This suggests a mechanism for the allosteric communication via the kinase and HEAT domains of VPS15”)*.

The last sentence of the abstract was modified per this suggestion. The speculations about the detailed mechanism of allostery in the discussion were consolidated into a single sentence, beginning with the phrase “It is tempting to speculate…”